# UNC-13L, UNC-13S, and Tomosyn form a protein code for fast and slow neurotransmitter release in *Caenorhabditis elegans*

**Zhitao Hu[1], Xia-Jing Tong[1], Joshua M Kaplan[1,2]***

[1]Department of Molecular Biology, Massachusetts General Hospital, Boston, United States; [2]Department of Neurobiology, Harvard Medical School, Boston, United States

**Abstract** Synaptic transmission consists of fast and slow components of neurotransmitter release. Here we show that these components are mediated by distinct exocytic proteins. The *Caenorhabditis elegans unc-13* gene is required for SV exocytosis, and encodes long and short isoforms (UNC-13L and S). Fast release was mediated by UNC-13L, whereas slow release required both UNC-13 proteins and was inhibited by Tomosyn. The spatial location of each protein correlated with its effect. Proteins adjacent to the dense projection mediated fast release, while those controlling slow release were more distal or diffuse. Two UNC-13L domains accelerated release. C2A, which binds RIM (a protein associated with calcium channels), anchored UNC-13 at active zones and shortened the latency of release. A calmodulin binding site accelerated release but had little effect on UNC-13's spatial localization. These results suggest that UNC-13L, UNC-13S, and Tomosyn form a molecular code that dictates the timing of neurotransmitter release.

**\*For correspondence:** kaplan@molbio.mgh.harvard.edu

**Competing interests:** The authors declare that no competing interests exist.

**Reviewing editor**: Graeme Davis, University of California, San Francisco, United States

## Introduction

The amount of neurotransmitter released at a synapse, and consequently synaptic strength, is often limited by the number of synapse vesicles (SVs) available for release. To become competent to undergo calcium-evoked fusion, SVs must first physically attach to the plasma membrane (termed docking), and then must undergo a second process that has been termed priming. Docking and priming are both promoted by UNC-13/Munc13 proteins, whereas Tomosyn inhibits both processes (*Verhage and Sorensen, 2008*). Mammals express four Munc13 isoforms (Munc13-1, ubMunc13-2, bMunc13-2, and Munc13-3), while *Caenorhabditis elegans* expresses two. It is not known if each Munc13 protein has distinct functions. Do they mediate different forms of plasticity? Do they promote exocytosis of different subpopulations of SVs?

At most synapses, neurotransmitter release comprises multiple populations of SVs that fuse with distinct kinetics and distinct release probabilities (*Neher and Sakaba, 2008*). These kinetic components of release are thought to be mediated by SVs in different spatial domains of the nerve terminal. Rapid (or synchronous) release occurs within a few milliseconds and is proposed to consist of fusion of SVs that are close to calcium entry sites. Delayed release occurs over tens to hundreds of milliseconds and is thought to be mediated by fusion of SVs that are farther from calcium channels.

Although a great deal is known about how synchronous and delayed release are regulated by calcium and by activity (*Zucker and Regehr, 2002*), relatively little is known about the molecules that define these kinetic components of release. It is generally believed that distinct calcium sensors are utilized for fast and slow release. Many studies suggest that the calcium sensor for fast release is Synaptotagmin I that binds calcium with low affinity and rapid kinetics (*Chapman, 2008*; *Pang and Sudhof, 2010*). A recent study proposed that the calcium sensor for slow release is Doc2α that binds

**eLife digest** Neurons communicate with one another at junctions called synapses. When an electrical signal known as an action potential travels along a neuron and arrives at a synapse, the neuron releases a package of transmitter chemicals into the synapse. These chemicals then diffuse across the gap and bind to receptors on a second neuron, conveying the signal to the target neuron.

The strength of a synapse depends in part on the number of packages, or vesicles, of transmitter chemicals that are available for release. Most synapses contain multiple populations of vesicles: some that are released within a few milliseconds of the arrival of an action potential, and others that are released more slowly. The vesicles that are released rapidly are found close to sites at which calcium ions enter the neuron, whereas the others are located further from these sites. However, little is known about the molecular basis of the differences between fast and slow vesicle release.

Now Hu et al. have studied the proteins involved in these two processes in *C. elegans*, a nematode worm that is often used in neuroscience because it has a simple nervous system, consisting of just 302 neurons, and a well-characterized genome. Hu et al. showed that the release of synaptic vesicles at the neuromuscular junction between neurons and muscles in *C. elegans* also has slow and fast components. A long form of UNC-13, which is also found in mammals, promotes fast release of transmitter vesicles. Slow release is mediated by an independent pathway that involves both long and short UNC-13 proteins, as well as a protein called Tomosyn. As in mammals, long UNC-13 is localized to the sites at which calcium ions enter neurons, whereas short UNC-13 is more widely distributed throughout neurons.

The work of Hu et al. provides a molecular explanation for how the timing of transmitter release is determined. Because the UNC-13 and Tomosyn proteins are evolutionarily conserved, this mechanism is likely to be present in other animals too.

calcium with high affinity and slow kinetics (*Yao et al., 2011*). Beyond the use of distinct calcium sensors, it is unclear if fast and slow release are distinguished by other exocytic proteins. Several important questions remain unanswered. Do SVs equilibrate between the slow and fast pools? Is there a molecular code that determines whether SVs enter the fast or slow pool? Are fast and slow SVs regulated by distinct mechanisms?

We utilized the *C. elegans* cholinergic neuromuscular junction (NMJ) as a model to address these questions. Transmission at this synapse is mediated by graded release of neurotransmitter, whereby release varies with the strength of depolarization (*Liu et al., 2009*). When activity is low, transmission consists of endogenous excitatory postsynaptic currents (EPSCs) that are evoked by fusion of a single SV (*Liu et al., 2005*). Following forced depolarization of the motor neuron (with a brief current pulse), large evoked EPSCs are produced consisting of the synchronous release of several hundred SVs. The ultrastructure of this synapse has been extensively characterized using the high-pressure freeze preservation technique. Each synapse contains a single dense projection that is surrounded by a pool of docked SVs, where docking occurs over a 1-μm domain of the presynaptic plasma membrane (*Weimer et al., 2006*; *Hammarlund et al., 2007*). In immunostained electron micrographs, UNC-2 calcium channels are enriched at dense projections, indicating that they are the sites of calcium entry (*Gracheva et al., 2008*). The active zone proteins UNC-10/RIM and UNC-13 promote docking of distinct population of SVs. In mutants lacking UNC-10/RIM, there was a reduction of docked SVs within 50 nm of the dense projection while SVs docked more distally were unaffected (*Weimer et al., 2006*; *Gracheva et al., 2008*). By contrast, in *unc-13* mutants, there was a uniform reduction of docked SVs across the entire 1-μm domain (*Gracheva et al., 2006*; *Weimer et al., 2006*; *Hammarlund et al., 2007*). Corresponding patterns of docking defects were observed in mouse Munc13 and RIM knockouts (*Siksou et al., 2009*; *Han et al., 2011*). UNC-10/RIM and UNC-13 immunostaining was also concentrated near dense projections in electron micrographs, consistent with their playing a relatively direct role in SV docking (*Weimer et al., 2006*). These results suggest that different proteins promote docking and fusion of distinct SV subpopulations. The functional significance of this spatial SV docking pattern remains unclear. Do proximal and distally docked SVs represent functionally distinct populations of SVs? Do they mediate different forms of neurotransmitter release or different forms of synaptic plasticity? Although this synapse has been characterized electrophysiologically, the mechanisms regulating release kinetics have not been analyzed.

Here we show that ACh release at this synapse consists of fast and slow components, and that these components are mediated by distinct SV priming factors. Fast release is mediated by exocytosis of SVs that are docked and primed by a long UNC-13 isoform (UNC-13L). Slow release is mediated by two UNC-13 isoforms (UNC-13L and S) and is inhibited by a third priming protein, the syntaxin-binding protein Tomosyn. These results suggest that UNC-13L, UNC-13S, and Tomosyn form a protein code that dictates the timing of neurotransmitter release.

## Results

### UNC-13L and S have distinct subcellular localization patterns

The *unc-13* gene contains two promoters. The upstream promoter drives expression of long isoforms, while a downstream promoter (which lies in the intron between exons 13 and 14) drives expression of short isoforms (hereafter referred to as UNC-13L and UNC-13S, respectively). UNC-13L and UNC-13S share a common domain (R, encoded by exons 15–31) that contains the C1, C2B, C2C, and MUN domains (*Figure 1A*; *Kohn et al., 2000*). UNC-13L proteins have an N-terminal domain (NTD, exons 1–13) containing the C2A domain, which the UNC-13S NTD (exon 14) lacks (*Figure 1A*). Rodents have analogous long (Munc13-1 and ubMunc13-2) and short (bMunc13-2 and Munc13-3) Munc13 proteins (*Brose et al., 2000*). In long Munc13 proteins, the C2A domain mediates formation of Munc13 homodimers and Munc13/RIM heterodimers (*Lu et al., 2006*). *C. elegans* UNC-13L (mCherry-tagged) and UNC-10 RIM (GFP-tagged) proteins were colocalized at presynaptic terminals (*Figure 1B–D*), consistent with immuno-EM studies indicating that they are concentrated near dense projections (*Weimer et al., 2006*). By contrast, UNC-13S had a diffuse distribution in axons and was less colocalized with UNC-10 RIM (*Figure 1B–D*). Their different localization patterns suggested that UNC-13L and UNC-13S may mediate different forms of release and, consequently, could have distinct effects on synaptic transmission.

### UNC-13L and UNC-13S both reconstitute synaptic function

To determine how each UNC-13 protein contributes to synaptic function, we constructed animals that selectively express one or the other isoform. For these experiments, we introduced transgenes that express either UNC-13S or UNC-13L into *unc-13(s69)* mutants. The *s69* allele corresponds to a 5 base pair deletion in an exon shared by UNC-13S and UNC-13L (exon 21), which shifts the reading frame thereby inactivating both isoforms (*Figure 1A*; *Kohn et al., 2000*). Hereafter, we refer to these transgenic strains as UNC-13S- and UNC-13L-rescued animals. Both UNC-13 transgenes rescued the locomotion rate defects of *unc-13(s69)* mutants (*Figure 2A,B*), indicating that both isoforms are able to restore synaptic function.

To more directly assess changes in synaptic transmission, we recorded EPSCs from adult body wall muscles. We analyzed endogenous EPSCs that are postsynaptic currents evoked by fusion of a single SV (*Liu et al., 2005*). Both UNC-13 transgenes rescued the endogenous EPSC defects of *unc-13(s69)* mutants (*Figure 2C,D*). Endogenous EPSC rates and amplitudes were restored to wild-type levels in UNC-13L rescue. UNC-13S rescue exhibited lower EPSC rates, and slightly smaller endogenous EPSC amplitudes than in wild-type controls. Thus, expressing either UNC-13L or UNC-13S reconstituted synaptic transmission. For both locomotion and endogenous EPSCs, UNC-13L produced better rescue than UNC-13S.

To confirm that UNC-13 transgenes rescued synaptic function by restoring SV priming, we measured the EPSC evoked by treatment with hypertonic sucrose (*Figure 2—figure supplement 1*). Hypertonic sucrose evokes release of docked and primed SVs in a calcium-independent manner (*Rosenmund and Stevens, 1996*). The sucrose-evoked charge observed in UNC-13L- and UNC-13S-rescued animals was significantly higher than in *unc-13(s69)* mutants, indicating that both isoforms efficiently promote SV priming. The sucrose-evoked charge observed in strains containing UNC-13S transgenes was significantly larger than those observed in wild-type controls, consistent with our prior studies (*Madison et al., 2005*) presumably because the primed pool of SVs is limited by UNC-13S expression levels.

### Fast- and slow-evoked ACh release are mediated by different UNC-13 proteins

To assess changes in the kinetics of ACh release, we recorded stimulus-evoked EPSCs (*Figure 3A*). To estimate the amount of ACh released during an evoked response, we calculated quantal content, which corresponds to the ratio of the total synaptic charge in an evoked EPSC to that in an endogenous

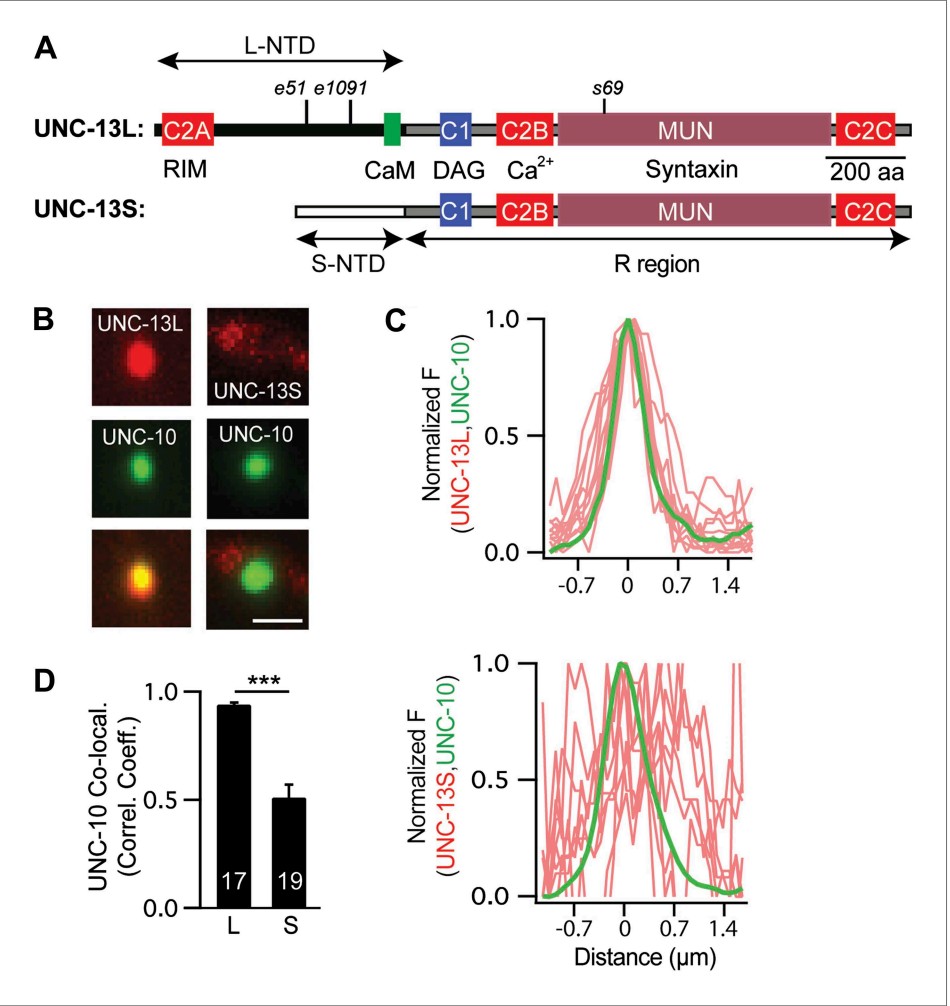

**Figure 1**. UNC-13L and UNC-13S have distinct localization patterns in axons. (**A**) The domain structure of UNC-13L (encoded by exons 1–31, excluding exon 14) and UNC-13S (exons 14–31) are illustrated. Ligands for each domain are indicated. The R region (1205 aa, comprising C1, C2B, MUN, and C2C domains) is shared between the two isoforms and is encoded by exons 15–31. Each isoform has a unique NTD. The L-NTD (610 aa) is encoded by exons 1–13 and contains a C2A domain (aa 1–96) and a predicted calmodulin binding site (green box, aa 556–610). The S-NTD (259 aa) is encoded by exon 14. The *e51* and *e1091* alleles correspond to nonsense mutations in L-NTD exons (exons 11 and 12, respectively). The *s69* allele corresponds to a 5 base pair deletion in an R domain exon (exon 21). (**B**–**D**) The localization of UNC-10/RIM in dorsal cord axons is compared to that of UNC-13L and UNC-13S. GFP-tagged UNC-10 and mCherry-tagged UNC-13L and UNC-13S were expressed in the DA and DB motor neurons of wild-type animals (using the *unc-129* promoter). Representative images (**B**), line scans (**C**) of active zones (identified by UNC-10/RIM fluorescence), and correlation coefficients (**D**) for UNC-10/RIM and UNC-13 fluorescence at synaptic puncta are shown. Line scans show normalized fluorescence values for UNC-10 (green) and UNC-13 (red). The UNC-10 trace is the averaged line scan from 20 puncta. The red traces represent UNC-13 line scans at 10 representative active zones. Scale bar indicates 1 µm. Values that differ significantly are indicated (***p<0.001). The number of animals analyzed is indicated for each genotype. Error bars indicate SEM.

EPSC (*Figure 3B*). Both transgenes partially rescued the *unc-13(s69)* quantal content defect (UNC-13L rescued 34% WT, UNC-13S rescued 86% WT, and *unc-13* mutant 0.5% WT).

Evoked EPSC kinetics in UNC-13L- and UNC-13S-rescued animals were significantly different (*Figure 3C,D*). In UNC-13L-rescued animals, evoked EPSCs exhibited a shorter lag between the stimulus and release (i.e., 0–20% rise time) as well as faster activation and decay kinetics than in wild-type controls, which resulted in a faster rate of charge transfer during evoked responses. Similarly, evoked responses in animals rescued with mCherry-tagged UNC-13L also exhibited faster activation and charge transfer

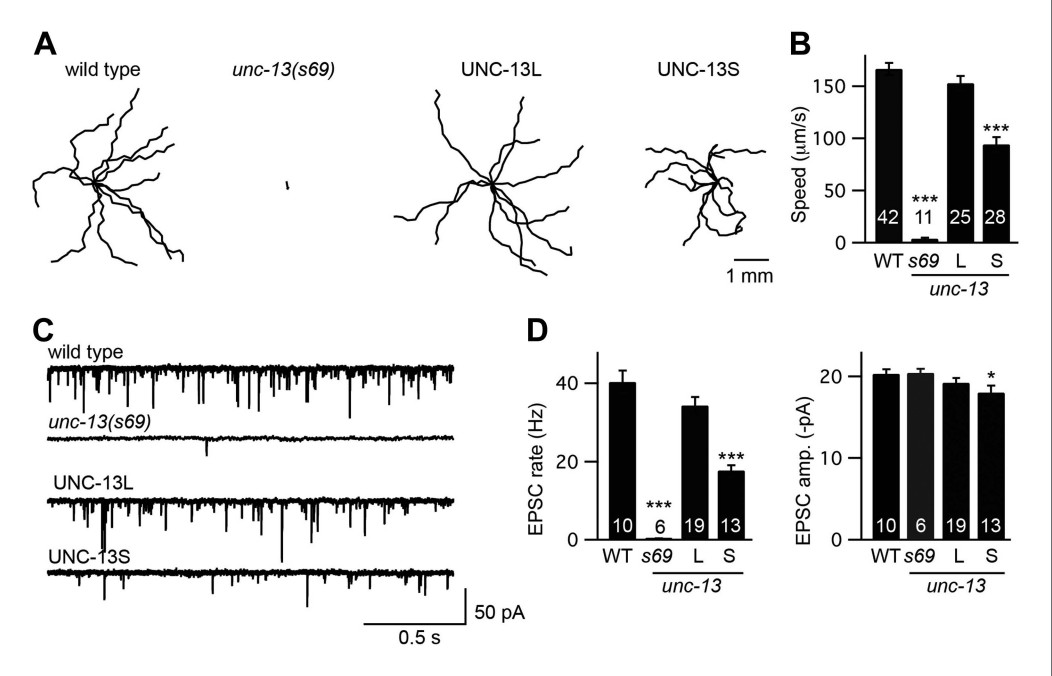

**Figure 2**. UNC-13L and UNC-13S both reconstitute locomotion behavior in *unc-13* mutants. (**A** and **B**) Locomotion behavior was analyzed for the indicated genotypes. (**A**) Representative 20 s locomotion trajectories are shown (n = 10 animals for each genotype). The starting points for each trajectory were aligned for clarity. (**B**) Locomotion rates are compared. UNC-13L (KP6893) and UNC-13S (KP6899) refer to *unc-13(s69)* mutants containing the indicated transgenes. (**C** and **D**) Endogenous EPSCs were recorded from adult body wall muscles of the indicated genotypes. Representative traces (**C**) and summary data (**D**) are shown. Values that differ significantly from wild-type controls are indicated (***p<0.001, *p<0.05). The number of animals analyzed is indicated for each genotype. Error bars indicate SEM.

The following figure supplements are available for figure 2:

**Figure supplement 1**. UNC-13L and UNC-13S both restore SV priming.

---

than in wild-type controls, indicating that fusion to mCherry did not disrupt UNC-13L function (*Figure 3—figure supplement 1A–C*). By contrast, UNC-13S-evoked EPSCs activated more slowly and exhibited slower charge transfer. Coexpression of both UNC-13 isoforms in *unc-13(s69)* produced evoked responses with kinetics similar to wild-type controls (*Figure 3—figure supplement 1D–F*). These results suggest that UNC-13L and UNC-13S mediate fast and slow ACh release, respectively.

The altered EPSC kinetics observed in UNC-13 rescue strains could result from changes in the kinetics of muscle responses to ACh. This possibility seems unlikely because the UNC-13 transgenes are not expressed in muscles. Nevertheless, we did two experiments to test this possibility (*Figure 3—figure supplement 1G–J*). First, the kinetics of endogenous EPSCs were unaltered in UNC-13L and UNC-13S rescue, indicating that the intrinsic kinetics of muscle ACh responses were not affected (*Figure 3—figure supplement 1G,H*). Second, we analyzed the contribution of postsynaptic receptor composition to evoked EPSC kinetics (*Figure 3—figure supplement 1I,J*). Body muscles express two classes of nicotinic ACh receptors, homomeric ACR-16 receptors and heteropentameric levamisole receptors (LevRs), which have distinct kinetic properties. ACR-16 receptors exhibit fast desensitization, whereas LevRs desensitize slowly (*Francis et al., 2005*; *Touroutine et al., 2005*). Evoked EPSC decay kinetics was slower in *acr-16* mutants and was unaltered in *unc-29* Lev mutants. Evoked EPSC activation kinetics were unaltered in both *acr-16* and *unc-29* Lev mutants (*Figure 3—figure supplement 1J*). Thus, changes in postsynaptic receptor composition cannot account for the altered evoked EPSC kinetics observed in UNC-13L- and UNC-13S-rescued animals. Instead, these results suggest that the altered evoked EPSC kinetics observed in UNC-13 transgenic animals are caused by changes in the kinetics of ACh release.

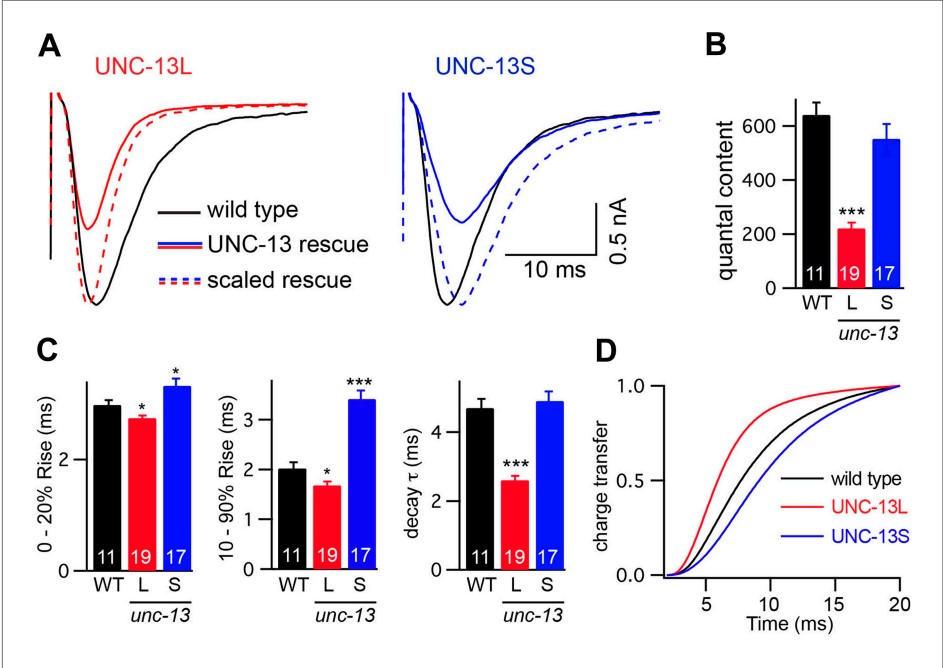

**Figure 3**. UNC-13L and UNC-13S mediate fast and slow release, respectively. Stimulus-evoked EPSCs were recorded from adults body wall muscles of the indicated genotypes. Averaged traces (**A**), quantal content (**B**), latency (0–20% rise time), activation (10–90% rise time), decay kinetics (**C**), and the cumulative charge transfer (**D**) are shown. Quantal content was calculated as the ratio of the charge transfer occurring during an evoked response to that occurring during an endogenous EPSC. Values that differ significantly from wild-type controls are indicated (***p<0.001, *p<0.05). The number of animals analyzed is indicated for each genotype. Error bars indicate SEM.

The following figure supplements are available for figure 3:

**Figure supplement 1**. Controls for experiments shown in **Figure 3**.

## Coupling between exocytosis and calcium entry dictates release kinetics

The lag (or coupling) between calcium entry and exocytosis is determined by how rapidly calcium reaches primed SVs and binds to their calcium sensors. Tighter coupling is expected when SVs are physically closer to calcium channels or when they have higher intrinsic calcium sensitivity. Thus, differences in evoked EPSCs kinetics could be mediated by differential coupling of primed SVs to the sites of calcium entry. To test this idea, we analyzed the effects of the calcium chelator EGTA on evoked responses. Because EGTA binds calcium slowly, it inhibits SV fusions that occur later in time, that is the SVs that are more loosely coupled with calcium entry (**Atluri and Regehr, 1996**).

To determine if fast and slow ACh release at the *C. elegans* NMJ can be distinguished by their calcium coupling, we analyzed the effect of EGTA on evoked EPSCs (**Figure 4**). We recorded evoked release after adding a membrane-permeant form of EGTA to the extracellular recording solution (40 µM EGTA-AM). EGTA-AM is loaded into both motor neurons and body muscles. In our experiments, body muscle EPSCs are recorded using patch pipettes containing 5 mM EGTA. Consequently, any effect of EGTA-AM on synaptic transmission must be caused by changes in presynaptic calcium. In wild-type animals, EGTA treatment significantly reduced the quantal content (56% decrease, p<0.001) of evoked EPSCs (**Figure 4**). EGTA treatment accelerated both the activation and decay of evoked EPSCs (**Figure 4—figure supplement 1A,B**), resulting in accelerated charge transfer during evoked responses (**Figure 4—figure supplement 1C**). EGTA had no effect on the latency of evoked release (**Figure 4—figure supplement 1C**). Thus, in wild-type controls, evoked release comprises two SV pools: a fast pool tightly coupled to calcium entry and a slow loosely coupled pool.

The effects of EGTA on EPSC kinetics could be caused by decreased presynaptic calcium levels or by faster decay of the presynaptic calcium transient. To distinguish between these possibilities,

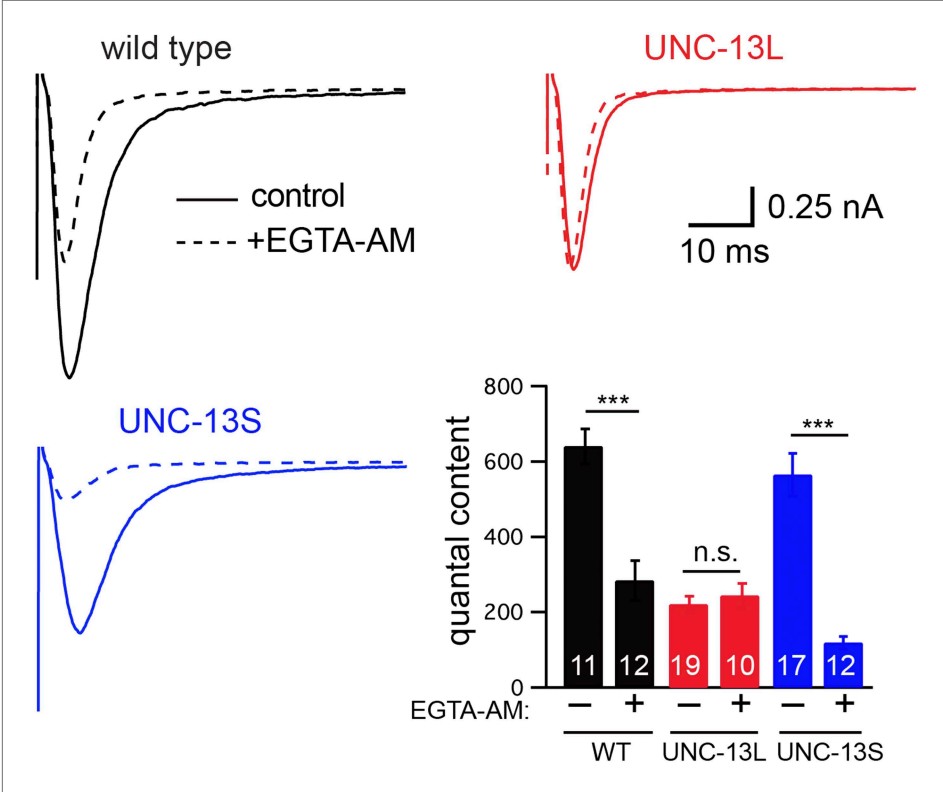

**Figure 4**. EGTA inhibits UNC-13S-mediated secretion but not that of UNC-13L. The effect of EGTA-AM on evoked responses was analyzed for the indicated genotypes. Averaged traces and quantal content are shown in control saline and after EGTA-AM treatment. Values that differ significantly are indicated (\*\*\*p<0.001; n.s., not significant). The number of animals analyzed is indicated for each genotype. Error bars indicate SEM.

The following figure supplements are available for figure 4:

**Figure supplement 1**. Changes in calcium influx cannot account for the altered evoked EPSC kinetics.

we analyzed the effect of external calcium concentration on evoked release (***Figure 4—figure supplement 1***). In reduced external calcium (0.5 mM $CaCl_2$), evoked EPSC amplitudes were very similar to those observed in 1 mM $CaCl_2$ with EGTA (***Figure 4—figure supplement 1A,B***). However, unlike EGTA treatment, reducing external calcium had no effect on EPSC activation and decay kinetics (***Figure 4—figure supplement 1B,C***). Thus, the effects of EGTA on EPSC kinetics are unlikely to be caused by a uniform decrease in presynaptic calcium and are more likely a consequence of accelerated decay of the presynaptic calcium transient, as previously proposed (***DiGregorio et al., 1999***).

## UNC-13L and UNC-13S have opposite effects on coupling of exocytosis to calcium entry

Differences in the evoked EPSC kinetics observed in UNC-13L- and UNC-13S-rescued animals could be mediated by altered coupling of primed SVs to calcium entry. Consistent with this idea, the effect of EGTA on the quantal content of evoked EPSCs was virtually eliminated in UNC-13L-rescued animals (***Figure 4***), indicating that SVs primed by UNC-13L bind calcium more rapidly than in wild-type controls. By contrast, in UNC-13S-rescued animals, the EGTA-resistant component of evoked responses was dramatically reduced (***Figure 4***), indicating that UNC-13S primed SVs bind calcium more slowly than in wild-type controls.

Collectively, these results suggest that different UNC-13 proteins mediate fast and slow release. UNC-13L is concentrated at active zones, promotes fusion of SVs that are tightly coupled to calcium entry, thereby mediating fast ACh release. UNC-13S is diffusely localized in axons, promotes fusion of SVs that are loosely coupled to calcium entry, thereby mediating slow ACh release.

## UNC-13L acts in conjunction with UNC-13S to promote slow ACh release

The preceding experiments suggest that UNC-13L and UNC-13S (when transgenically expressed) can function independently to promote fast and slow release. Because transgenes are typically expressed at higher levels than endogenous genes, we wanted to determine if endogenously expressed UNC-13 isoforms also function independently. To address this question, we analyzed *unc-13* mutants containing mutations that selectively inactivate UNC-13L. The *e51* and *e1091* alleles correspond to nonsense mutations in exons 11 and 12 (respectively) of *unc-13* (*Kohn et al., 2000*). These mutations truncate UNC-13L prior to the R domain (encoded by exons 15–31) (*Figure 1A*). The *e51* and *e1091* mutations are not included in the *unc-13S* mRNA (encoded by exons 14–31) (*Kohn et al., 2000*). Thus, *e51* and *e1091* both inactivate UNC-13L while leaving UNC-13S intact. Hereafter, we refer to *e51* and *e1091* as *unc-13* [L⁻S⁺] mutations.

If endogenous UNC-13S functions independently of UNC-13L, synaptic transmission in *unc-13* [L⁻S⁺] mutants should be similar to that observed in UNC-13S-rescued animals. Contrary to this idea, the evoked EPSC (*Figure 5A,B*) and endogenous EPSC (*Figure 5—figure supplement 1A,B*) defects observed in *unc-13* [L⁻S⁺] mutants were dramatically stronger than in UNC-13S rescue, but were slightly less severe than in *unc-13(s69)* mutants, consistent with prior studies (*Richmond et al., 1999*). Thus, unlike UNC-13S-rescued animals, mutants lacking endogenous UNC-13L had severe synaptic defects. To confirm that the transmission defect observed in *unc-13* [L⁻S⁺] was caused by decreased SV priming, we analyzed the sucrose-evoked EPSC. The sucrose-evoked charge was also dramatically reduced in *unc-13(e1091)* [L⁻S⁺] (22% WT) but was slightly larger than in *unc-13(s69)* mutants (10% WT, p<0.05) (*Figure 5—figure supplement 1C,D*) (*McEwen et al., 2006*). Thus, inactivating UNC-13L decreased the priming activities of both UNC-13S and UNC-13L.

The *unc-13* [L⁻S⁺]-evoked EPSC defect was fully rescued by a transgene expressing UNC-13L (*Figure 5A,B*), indicating that this defect was caused by inactivation of UNC-13L. The quantal content observed in UNC-13L rescued *unc-13* [L⁻S⁺] mutants was indistinguishable from that in wild-type controls and was significantly larger than in UNC-13L-rescued *unc-13(s69)* mutants (*Figure 5B*). The activation and decay kinetics of evoked EPSCs in UNC-13L-rescued *unc-13(e1091)* [L⁻S⁺] mutants were indistinguishable from wild-type controls (*Figure 5A,B*). EGTA treatment decreased the amplitude and quantal content of evoked EPSCs in UNC-13L-rescued *unc-13(e1091)* [L⁻S⁺] mutants (*Figure 5C,D*). Thus, restoring UNC-13L expression in *unc-13* [L⁻S⁺] mutants reconstituted both fast and slow release. By contrast, expressing UNC-13L in *unc-13(s69)* mutants reconstituted only fast release. These results suggest that endogenous UNC-13S requires UNC-13L to perform its function.

The discrepancy between the UNC-13S rescue and *unc-13* [L⁻S⁺] mutant phenotypes could result from a failure to express UNC-13S in *unc-13* [L⁻S⁺] mutants. To test this idea, we measured the abundance of UNC-13S mRNA by quantitative PCR. We found that UNC-13S mRNA levels were modestly increased in *unc-13* [L⁻S⁺] mutants compared to wild-type controls (38% increase, p<0.01). Consequently, decreased expression of the UNC-13S promoter is unlikely to explain the synaptic transmission defects observed in mutants lacking UNC-13L.

## Tomosyn inhibits slow release

Another potential model to explain the *unc-13* [L⁻S⁺] synaptic defect is that UNC-13S function is actively inhibited by an endogenous regulatory mechanism. In this scenario, overexpressing UNC-13S could titrate out the inhibitor (through direct binding or indirectly via a shared binding partner), thereby restoring synaptic function. Several studies suggest that the Tomosyn protein could provide this inhibitory function. In mutants lacking Tomosyn, neurotransmitter release is increased (*Gracheva et al., 2006*; *McEwen et al., 2006*). In electron micrographs, anti-TOM-1 antibodies labeled a presynaptic domain that is distal to dense projections, and little staining was observed adjacent to dense projections (*Gracheva et al., 2007*). In *tom-1* Tomosyn mutants, docking of distal SVs was significantly increased (*Gracheva et al., 2006*).

Prompted by these results, we tested the idea that TOM-1 specifically inhibits slow ACh release during evoked responses. Analysis of the kinetics of evoked responses suggests that ACh was released more slowly in *tom-1* mutants. In particular, evoked EPSC decay was significantly slower in *tom-1* mutants (*Figure 6*). The altered kinetics of *tom-1*-evoked responses are unlikely to result from a change in the intrinsic kinetics of muscle ACh responses because endogenous EPSC kinetics in *tom-1* mutants were indistinguishable from those in wild-type controls (*Figure 3—figure supplement 1G,H*).

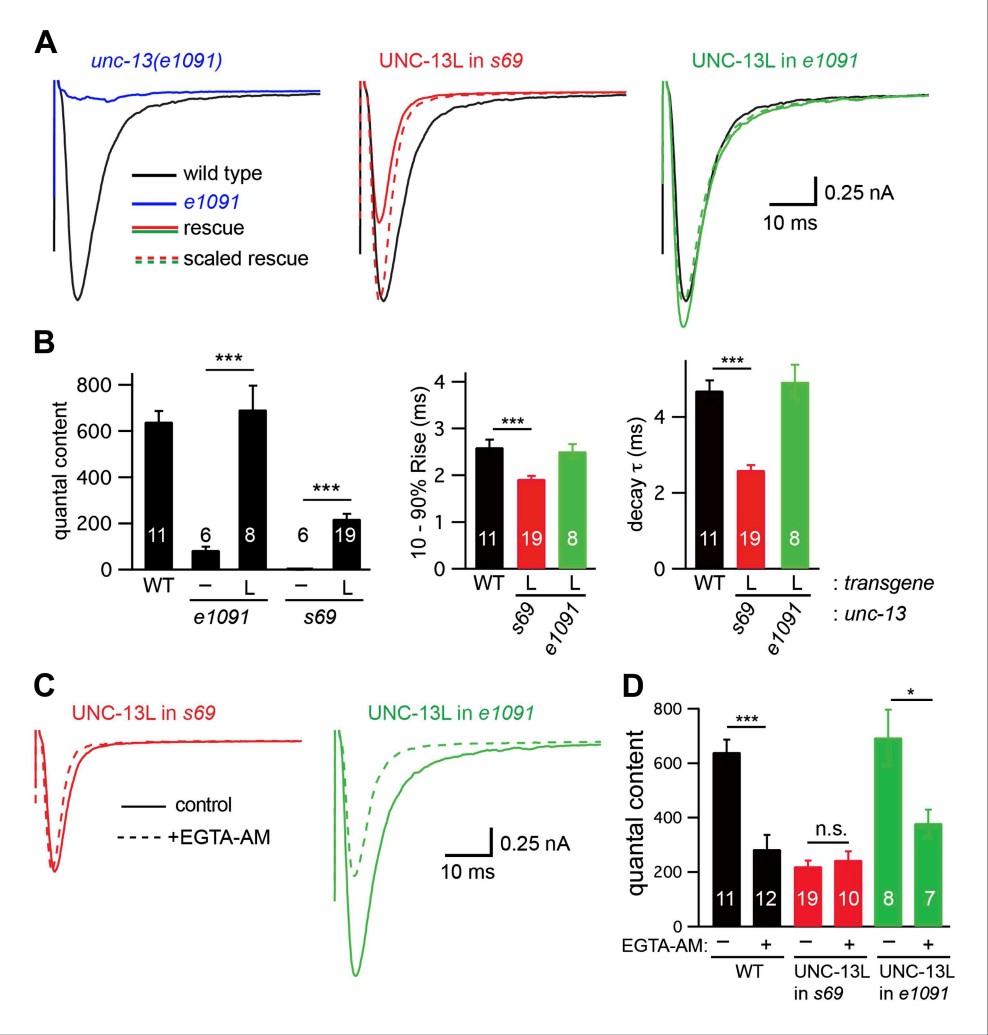

**Figure 5**. UNC-13L is required for the priming activity of endogenous UNC-13S. Stimulus-evoked EPSCs (**A** and **B**) were recorded from adult body wall muscles of the indicated genotypes. Averaged responses (**A**), and summary data (**B**) for quantal content, activation and decay kinetics are shown. (**C** and **D**) Expression of UNC-13L in *unc-13(e1091)* [L⁻S⁺] mutants restores both EGTA sensitive and resistant forms of release. By contrast, UNC-13L expression in *unc-13(s69)* reconstitutes only the EGTA-resistant component of release. Evoked EPSCs were recorded in control saline and after EGTA-AM treatment. Averaged responses (**C**) and quantal content (**D**) are shown. Values that differ significantly are indicated (***p<0.001; *p<0.05; n.s., not significant). The number of animals analyzed is indicated for each genotype. Error bars indicate SEM.

The following figure supplements are available for figure 5:

**Figure supplement 1**. UNC-13L is required for the priming activity of endogenously expressed UNC-13S.

---

The increased evoked EPSC amplitude and quantal content observed in *tom-1* mutants were both eliminated by EGTA treatment, indicating that the increased ACh release was mediated by fusion of SVs that are loosely coupled to calcium entry (**Figure 6**). Collectively, these results suggest that evoked ACh release was slower and more prolonged in *tom-1* mutants, and that this kinetic change was mediated by increased exocytosis of SVs that are docked distal to the site of calcium entry. Thus, TOM-1 potently inhibits slow ACh release and has more limited effects on fast release.

## Tomosyn inhibits UNC-13S-mediated secretion

To determine if TOM-1 inhibits UNC-13S function, we analyzed *tom-1 unc-13* double mutants (**Figure 7**). As previously reported, the quantal content of evoked EPSCs (**Figure 7B**) and the total sucrose-evoked

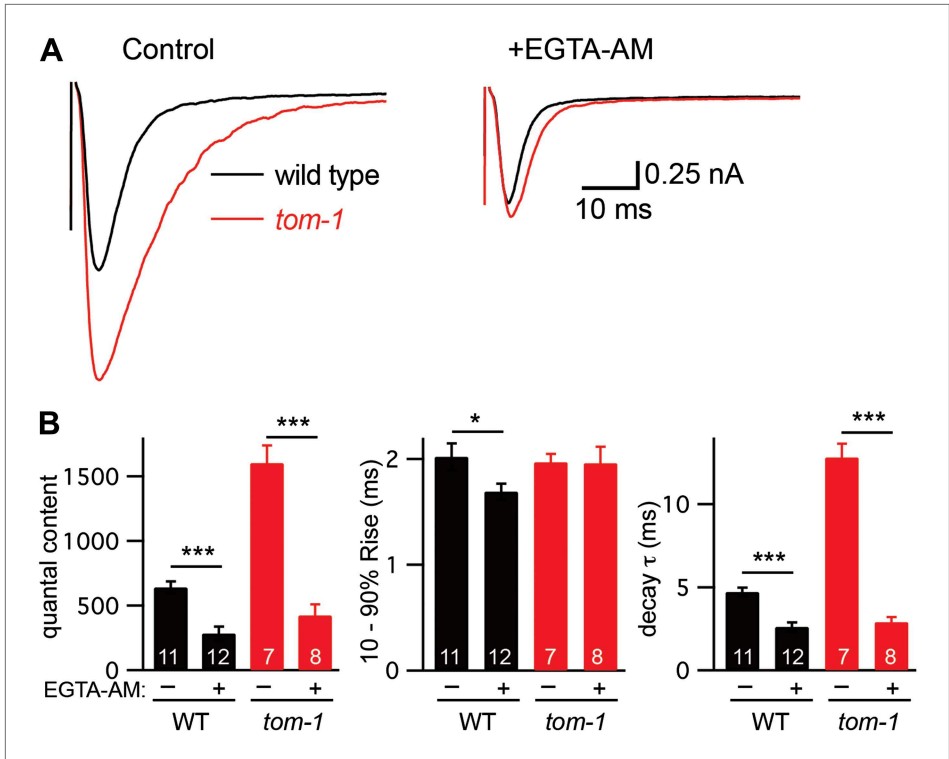

**Figure 6**. Tomosyn inhibits slow release. Stimulus-evoked EPSCs were recorded from body wall muscles in control saline and after the addition of EGTA-AM. Averaged traces (**A**) and summary data (**B**) for quantal content, activation, and decay kinetics are shown. Values that differ significantly are indicated (***p<0.001; n.s., not significant). The number of animals analyzed is indicated for each genotype. Error bars indicate SEM.

charge (**Figure 7—figure supplement 1A,B**) were significantly increased in *tom-1* single mutants, indicating increased SV priming (**Gracheva et al., 2006**; **McEwen et al., 2006**). Evoked ACh release was nearly completely eliminated in *tom-1 unc-13(s69)* double mutants (**Figure 7A,B**), implying that UNC-13 priming activity is required for TOM-1's effects on ACh release. By contrast, evoked ACh release in *tom-1 unc-13* [L⁻S⁺] double mutants was significantly larger than in *unc-13* [L⁻S⁺] single mutants (**Figure 7A,B**). Increased ACh release in *tom-1 unc-13* [L⁻S⁺] double mutants was accompanied by a corresponding increase in the sucrose-evoked charge (**Figure 7—figure supplement 1A,B**), implying that TOM-1 inhibits UNC-13S priming activity. Consistent with improved synaptic function, the locomotion rate of *tom-1 unc-13* [L⁻S⁺] double mutants was significantly higher than in *unc-13* [L⁻S⁺] single mutants (**Figure 7C,D**). Thus, electrophysiological recordings and behavioral studies indicate that synaptic function in *tom-1 unc-13* [L⁻S⁺] double mutants was significantly greater than in *tom-1 unc-13(s69)* double mutants. The only genetic difference between these strains is that *tom-1 unc-13* [L⁻S⁺] double mutants retain endogenous UNC-13S, while *tom-1 unc-13(s69)* double mutants lack UNC-13S. Consequently, these results suggest that Tomosyn inhibits the priming activity of endogenously expressed UNC-13S.

To determine if Tomosyn also inhibits UNC-13L, we analyzed synaptic transmission in UNC-13S- and UNC-13L-rescued animals following inactivation of Tomosyn (**Figure 7—figure supplement 1C,D**). Inactivating Tomosyn in UNC-13S animals significantly increased the quantal content (263% increase, p<0.001). By contrast, inactivating Tomosyn in UNC-13L-rescued animals had more modest effects on the quantal content (96% increase, p<0.001). These results suggest that Tomosyn potently inhibits UNC-13S-mediated secretion and has weaker effects on UNC-13L. As previously reported (**Gracheva et al., 2006**; **McEwen et al., 2006**), ACh release and SV priming were slightly higher in *tom-1 unc-13(s69)* double mutants than in *unc-13(s69)* single mutants (**Figure 7—figure supplement 1**). These results suggest that Tomosyn also inhibits an UNC-13-independent form of release.

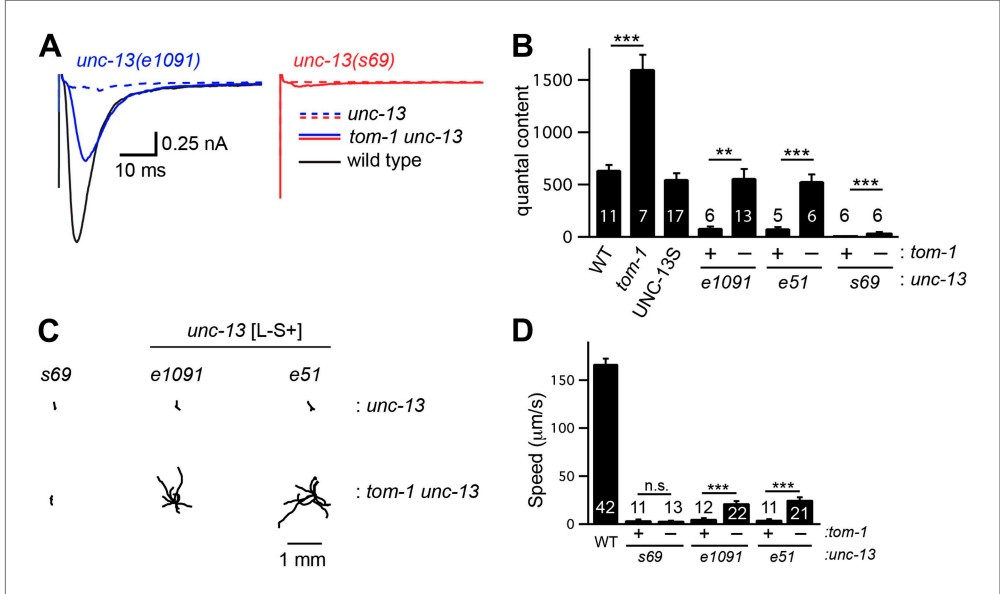

**Figure 7**. Tomosyn inhibits UNC-13S-mediated secretion. Inactivating TOM-1 Tomosyn in *unc-13* [L⁻S⁺] (*e1091*) mutants restored ACh release (**A** and **B**) and locomotion behavior (**C** and **D**). (**A** and **B**) Stimulus-evoked EPSCs were analyzed for the indicated genotypes. Averaged traces (**A**) and quantal content (**B**) are shown. (**C** and **D**) Locomotion behavior was analyzed in the indicated strains. Representative 20 s locomotion trajectories are shown (n = 10 animals for each genotype) (**C**). The starting points for each trajectory were aligned for clarity. (**D**) Locomotion rates are compared for the indicated genotypes. Values that differ significantly are indicated (***p<0.001; **p<0.01; n.s., not significant). The number of animals analyzed is indicated for each genotype. Error bars indicate SEM.

The following figure supplements are available for figure 7:

**Figure supplement 1**. TOM-1 inhibits the priming activity of UNC-13S.

**Figure supplement 2**. The latency of evoked responses is prolonged and calcium coupling is loosened in *unc-13(e1091)* [L⁻S⁺] and in *unc-10* RIM mutants.

## UNC-13L and UNC-10/RIM are required for fast release

Thus far, our results suggest that fast release is mediated by UNC-13L, which is concentrated at dense projections, and by fusion of SVs that are tightly coupled to calcium channels. These results suggest that mutants lacking UNC-13L, or its binding partner UNC-10/RIM, should have slower release. To test this idea, we compared release kinetics in *unc-13(e1091)* [L⁻S⁺] and *unc-10* RIM mutants. Because both mutations dramatically reduce release, we analyzed double mutants lacking TOM-1/Tomosyn to increase the sensitivity of our experiments. The activation kinetics of evoked EPSCs in *tom-1 unc-13(e1091)* and in *tom-1; unc-10* double mutants were significantly slower than those in *tom-1* single mutants (***Figure 7—figure supplement 2A,B***). In particular, the lag between the stimulus and activation of evoked release was prolonged in both double mutants (***Figure 7—figure supplement 2B***). Evoked responses in *tom-1 unc-13(e1091)* and in *tom-1; unc-10* double mutants were also significantly less EGTA-resistant than those in *tom-1* single mutants (***Figure 7—figure supplement 2C,E***). Collectively, these results suggest that UNC-13L and UNC-10/RIM are required for fast release. The role of RIM in promoting fast release is likely conserved, as mouse and fly RIM knockouts exhibit similar changes in release kinetics and calcium coupling (***Han et al., 2011***; ***Müller et al., 2012***). Mutants lacking RIM also have decreased calcium channel density and decreased calcium entry at active zones (***Han et al., 2011***; ***Kaeser et al., 2011***; ***Müller et al., 2012***). Therefore, *unc-10* RIM mutations may alter the spatial and temporal extent of the calcium transient, which would contribute to changes in release kinetics.

## The C2A domain recruits UNC-13L to active zones but is not required for fast release kinetics

The two UNC-13 isoforms differ only in their NTDs (*Figure 1A*). To test the functional importance of the NTDs, we expressed a truncated UNC-13 protein containing only the common region (UNC-13R) (*Figure 8—figure supplement 1*). UNC-13R had a diffuse distribution in axons (*Figure 8A–C*), and produced evoked responses with activation kinetics and EGTA sensitivity indistinguishable from those in UNC-13S rescue (*Figure 8D–H*). Thus, sequences in the UNC-13S NTD were not required for the spatial location, looser calcium coupling, and slower release kinetics of UNC-13S.

Faster kinetics and tighter calcium coupling could be caused by the closer spatial relationship of UNC-13L to the dense projection (*Figure 1B–D*) (*Weimer et al., 2006*). To test this idea, we analyzed a chimeric protein containing the C2A domain (which binds RIM) fused to the R domain (C2AR) (*Figure 8—figure supplement 1*). As expected, mCherry-tagged C2AR was more punctate, and exhibited greater colocalization with UNC-10/RIM than did mCherry-tagged UNC-13R (*Figure 8A–C*). Thus, the C2A domain increased UNC-13 recruitment to active zones. Evoked EPSCs in C2AR had slightly shorter latencies than those in UNC-13R; however, the EGTA resistance of evoked responses in C2AR and UNC-13R were similar (*Figure 8D–H*). Thus, fusing the C2A and R domains recapitulated the spatial localization of UNC-13L but had no effect on calcium coupling and only a modest effect on release kinetics. These results suggest that differences in the spatial localization of UNC-13S and L cannot fully account for differences in the kinetics of neurotransmitter release mediated by these proteins.

To further test C2A's role in release kinetics, we analyzed an UNC-13L deletion mutant lacking the C2A domain (ΔC2A) (*Figure 8—figure supplement 1*). UNC-10/RIM was significantly more colocalized with ΔC2A than with UNC-13R (*Figure 8A–C*). ΔC2A evoked EPSCs had shorter latencies, faster activation kinetics, and greater EGTA resistance than those in UNC-13R-rescued animals (*Figure 8D–H*). Thus, the C2A domain was not required for UNC-13 anchoring at active zones, for tight coupling of exocytosis to calcium entry, nor for rapid release kinetics.

## UNC-13L's calmodulin binding site accelerates release

Our analysis of ΔC2A suggests that in addition to the C2A domain, other sequences in the UNC-13L NTD contribute to the faster release kinetics. The NTDs of Munc13 proteins have amphipathic helixes that bind calmodulin (*Junge et al., 2004*; *Lipstein et al., 2012*). Calmodulin binds to Munc13 in an extended conformation, where the calmodulin C-terminal lobe is anchored to hydrophobic residues at positions 1, 5, and 8/10 of the Munc13 amphipathic helix, while the N-terminal lobe is anchored at position 26 of the helix (*Rodriguez-Castaneda et al., 2010*; *Lipstein et al., 2012*). A similar amphipathic helix is present at the same location in the UNC-13L NTD (*Figure 9A*).

To test its functional importance, we constructed a chimeric UNC-13 protein containing the predicted calmodulin-binding site (residues 556–610) fused to the R domain (CaMR) (*Figure 8—figure supplement 1*). An mCherry-tagged CaMR had a diffuse distribution in axons and was not colocalized with UNC-10/RIM, similar to the distribution of UNC-13R (*Figure 9B–D*). Thus, the CaM-binding site had little effect on UNC-13's spatial distribution. CaMR-mediated evoked responses had significantly faster activation kinetics and were significantly more EGTA resistant than those in UNC-13R-rescued animals (*Figure 9E–I*). Both these effects were eliminated by a mutation predicted to disrupt calmodulin binding, CaMR[F574R] (*Junge et al., 2004*). In particular, CaMR-evoked responses exhibited faster charge transfer, shorter latency (i.e., 0–20% rise time), and greater EGTA resistance than those in CaMR[F574R] (*Figure 9F–I*). Collectively, these results suggest that the UNC-13L NTD has two effects on release. The C2A domain anchors UNC-13 at active zones, while the calmodulin binding site domain accelerates release kinetics and tightens calcium coupling.

## Discussion

Here we show that synaptic transmission at the *C. elegans* NMJs is a composite of fast and slow ACh release, and we describe a molecular mechanism that dictates release kinetics. Fast release comprises fusion of SVs that are tightly coupled to calcium entry, and is mediated by UNC-13L. Slow release consists of SV fusions that are loosely coupled to calcium entry is promoted by UNC-13S and L acting in concert, and is inhibited to TOM-1 Tomosyn. UNC-13L accelerates release by two mechanisms: increasing the spatial proximity of primed SVs to calcium channels and enhancing the calcium sensitivity of release by binding to calmodulin. These experiments define a molecular code that dictates the

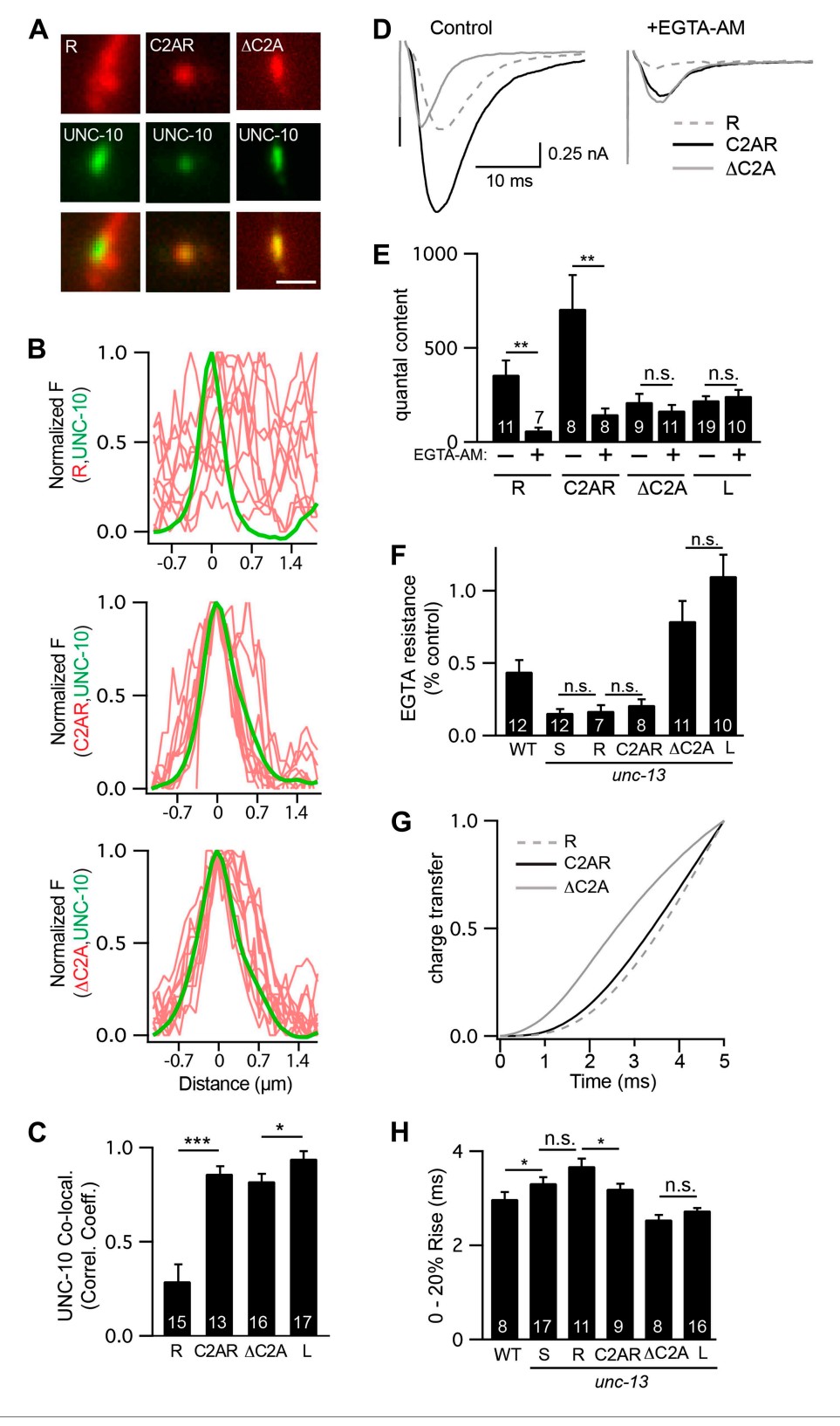

**Figure 8**. The C2A domain anchors UNC-13 at active zones and shortens the latency of release. Fusing the C2A domain to the R region (C2AR) increased recruitment of UNC-10 to active zones, shortened the latency of release, but did not tighten the coupling of release to calcium entry. Deleting the C2A domain in full length UNC-13L
*Figure 8. Continued on next page*

*Figure 8. Continued*

(ΔC2A) modestly decreased targeting to active zones, did not lengthen the latency of release, and did not loosen the coupling of exocytosis and calcium entry. (**A** and **C**) The distribution of mCherry-tagged UNC-13R, C2AR, and ΔC2A at active zones (labeled with GFP-tagged UNC-10/RIM) are shown. All tagged proteins were expressed in DA and DB motor neurons of wild type animals (using the *unc-129* promoter). Representative images (**A**), line scans (**B**), and correlation coefficients (**C**) for UNC-10/RIM and UNC-13 fluorescence at synaptic puncta are shown. Scale bar indicates 1 μm. (**D**–**H**) Stimulus-evoked EPSCs were recorded from body wall muscles in control saline and after the addition of EGTA-AM. Averaged traces (**D**) and summary data for quantal content (**E**), EGTA resistance (**F**), time course of initial evoked charge transfer (0–5 ms, **G**), and the latency of release (0–20% rise time, **H**) are shown. Values that differ significantly are indicated (***p<0.001; **p<0.01; *p<0.05; n.s., not significant). The number of animals analyzed is indicated for each genotype. Error bars indicate SEM.
The following figure supplements are available for figure 8:
**Figure supplement 1**. Chimeric UNC-13 proteins utilized in **Figures 8 and 9**.

kinetics of neurotransmitter release during evoked responses. Below we discuss the significance of these findings.

## Tomosyn inhibits slow release

Prior studies showed that TOM-1 inhibits SV priming, and that it does so by binding to plasma membrane SNAREs, forming inactive SNARE complexes (*Gracheva et al., 2006*; *McEwen et al., 2006*). Here we show that Tomosyn's effect on synaptic transmission is mediated primarily by inhibiting UNC-13S-mediated release. The increased evoked ACh release in *tom-1* mutants was eliminated by EGTA treatment, indicating that the extra release was mediated by fusion of SVs that are loosely coupled to calcium entry. In transgenic animals, TOM-1 inhibited UNC-13S-mediated slow release far more potently than UNC-13L-mediated fast release. The apparent selectivity for inhibiting slow release could result from the distal localization of the TOM-1 protein, as immuno-EM studies indicate that TOM-1 staining peaks ~300 nm from the dense projection (*Gracheva et al., 2007*). We previously reported that the synaptic abundance of UNC-13S is increased in *tom-1* mutants, perhaps because the rate of distal SV fusions had increased (*McEwen et al., 2006*). Taken together, these results strongly support the idea that Tomosyn inhibits priming and fusion of distal SVs.

## Spatial differences between fast and slow release

Several results suggest that fast and slow release were mediated by fusion of SVs that are docked to distinct spatial domains of the nerve terminal. Immuno-EM studies indicate that proteins mediating fast release (i.e., UNC-10/RIM and UNC-13L) are localized near dense projections (*Weimer et al., 2006*; *Gracheva et al., 2008*). Slow release is controlled by UNC-13S, which has a diffuse localization, and by TOM-1 for which immuno-EM studies indicate a distal location (*Gracheva et al., 2007*).

Analysis of SV docking by high-pressure freeze preservation and electron microscopy also suggests that fast and slow release are mediated by SV fusions in distinct spatial domains. Each NMJ has a single dense projection surrounded by 38 docked SVs extending radially (*Hammarlund et al., 2007*). A mutation inactivating both UNC-13 isoforms decreased SV docking across the entire active zone (i.e., <350 nm from the dense projection) (*Gracheva et al., 2006*; *Weimer et al., 2006*; *Hammarlund et al., 2007*). By contrast, docking of proximal SVs was significantly reduced in *unc-13(e1091)* [L⁻S⁺] mutants (<100 nm from dense projections) and in *unc-10* RIM mutants (<50 nm from dense projections), whereas docking of distal SVs was unaffected in both mutants (*Weimer et al., 2006*; *Hammarlund et al., 2007*; *Gracheva et al., 2008*). We found that *unc-13(e1091)* [L⁻S⁺] and *unc-10* RIM mutations made evoked release slower and more loosely coupled to calcium entry, both indicating that SVs docked proximal to the dense projection are required for fast release. By contrast, inactivating TOM-1/Tomosyn increased docking of distal SVs (*Gracheva et al., 2006*) and made ACh release slower and more loosely coupled to calcium entry. Finally, the proportion of docked SVs that are <100 nm from the dense projection (15/38, or ~40%) (*Hammarlund et al., 2007*) is similar to the fraction of evoked release that was EGTA resistant in our conditions.

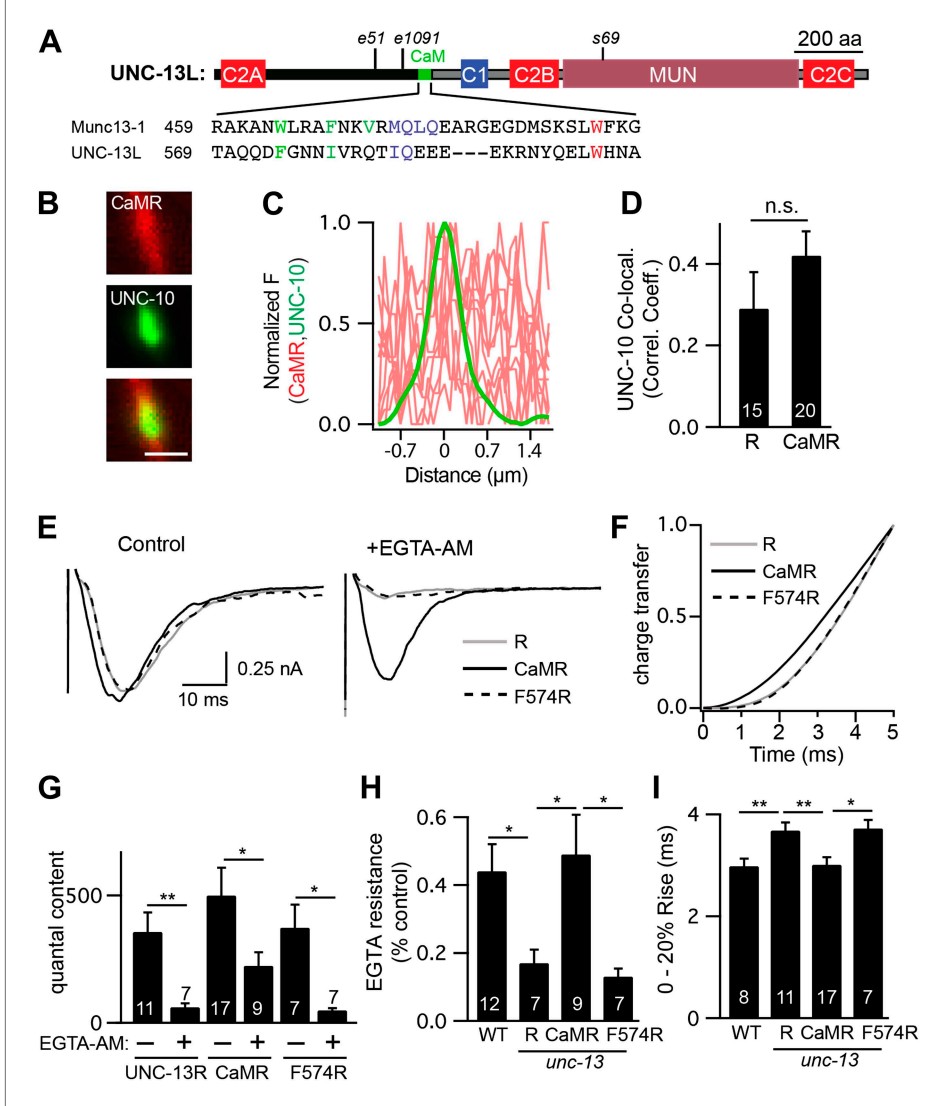

**Figure 9**. A predicted calmodulin-binding site in UNC-13L accelerates release. The predicted calmodulin (CaM)-binding site of UNC-13L (residues 556–610) was fused to the R domain. CaMR had a diffuse axonal distribution, was poorly colocalized with UNC-10/RIM, but exhibited faster and more EGTA-resistant evoked ACh release. The effects of the CaM-binding site on release were eliminated by the F574R mutation, which is predicted to disrupt calmodulin binding (**Junge et al., 2004**). (**A**) Position of the putative CaM-binding site in UNC-13L and its alignment with the rat Munc13-1 CaM-binding site sequence are shown. Hydrophobic residues anchoring the C-terminal (green) and N-terminal (red) lobes of CaM, and predicted apo-calmodulin binding sites (blue) are indicated (**Rodriguez-Castaneda et al., 2010**; **Lipstein et al., 2012**). (**B–D**) The distribution of mCherry-tagged CaMR at active zones (labeled with GFP-tagged UNC-10/RIM) are shown. Tagged proteins were expressed in DA and DB motor neurons of wild-type animals (using the *unc-129* promoter). Representative images (**B**), line scans (**C**), and correlation coefficients (**D**) for UNC-10/RIM and CaMR fluorescence at synaptic puncta are shown. Scale bar indicates 1 µm. (**E** and **H**) Stimulus-evoked EPSCs were recorded from body wall muscles in control saline and after the addition of EGTA-AM. Averaged traces (**E**) and summary data for time course of initial evoked charge transfer (0–5 ms, **F**), quantal content (**G**), EGTA resistance (**H**), and the latency of release (0–20% rise time, **I**) are shown. Values that differ significantly are indicated (**p<0.01; *p<0.05; n.s., not significant). The number of animals analyzed is indicated for each genotype. Error bars indicate SEM.

Thus, electrophysiological recordings, ultrastructural analysis of SV docking, immuno-EM localization of priming proteins (UNC-10/RIM, UNC-13L, and TOM-1), and sensitivity to EGTA inhibition all support a model whereby fast ACh release is mediated by the subpopulation of SVs that are docked

adjacent to dense projections (<100 nm), while slow release is mediated by fusion of distal SVs (100–350 nm from dense projections).

## UNC-13L enhances release by three mechanisms

Are differences in the spatial location of primed vesicles sufficient to explain the kinetics of fast and slow release? Several results argue against this idea. The rise time of fast and slow release differ by ~2 ms. If this difference was caused solely by a diffusional delay in calcium binding, slow SVs would have to be 3 µm farther from calcium channels than fast SVs (assuming $D_{ca}$ = 200 µm$^2$/s). Thus, it seems likely that exocytosis of UNC-13S primed SVs is intrinsically slower than that of UNC-13L primed vesicles.

We show that UNC-13L accelerates release by two mechanisms. The C2A domain increased UNC-13 anchoring at active zones, and produced a slight decrease in the delay between stimulation and the onset of release. Both effects support the idea that the C2A domain caused primed SVs to become physically closer to calcium channels, thereby decreasing the time required for calcium to reach the vesicle's calcium sensor. Other UNC-13L sequences (beyond C2A) also contribute to its physical location because the ΔC2A protein was strongly colocalized with UNC-10/RIM. The calmodulin-binding site in UNC-13L's NTD also accelerated release. The analogous calmodulin-binding site in mouse Munc13-1 and Munc13-2 is required for augmentation, a form of short-term plasticity (*Junge et al., 2004*). Based on these results, it was proposed that calmodulin binding to Munc13 comprises a calcium sensor for short-term plasticity. Our results suggest that this calmodulin-binding site also accelerates release. Thus, the faster kinetics of UNC-13L-mediated release results from both tighter spatial coupling to calcium channels and faster intrinsic kinetics of exocytosis.

In addition to promoting fast release, UNC-13L also acts in conjunction with UNC-13S to promote slow release. Mutations inactivating UNC-13L dramatically reduced fast and slow release, and both forms of release were re-instated by restoring UNC-13L expression. Thus, UNC-13L promotes but is not required for slow release. The UNC-13L requirement for slow release was reduced (but not eliminated) when the total number of primed SVs was increased (by inactivating Tomosyn or by overexpressing UNC-13S). UNC-13L could regulate slow release by a variety of mechanisms. UNC-13S mRNA levels were not decreased in *unc-13* [L⁻S⁺] mutants, implying that UNC-13L did not alter expression of the UNC-13S promoter. UNC-13L could directly bind to UNC-13S thereby altering its activity or stability. UNC-13L could prevent inhibition of UNC-13S by Tomosyn. Finally, UNC-13L could increase the release probability of UNC-13S primed SVs. Further experiments will be required to determine how UNC-13L regulates slow release.

Our results do not exclude the idea that other potential mechanisms also contribute to differences in UNC-13L and UNC-13S release kinetics. For example, UNC-13 and TOM-1 could regulate the kinetics or duration of calcium entry or could alter the spatial distribution of open calcium channels. Further experiments will be required to test these ideas.

## Implications for understanding synaptic plasticity and circuit function

Our results suggest that fast and slow release are mediated by exocytosis of distinct subpopulations of SVs, and that distinct priming molecules mediate fast and slow SV fusions. In this manner, fast and slow release could be independently adjusted by modulatory pathways. It is intriguing that the fast and slow release components appear to function in parallel. We were able to construct mutants in which evoked release is mediated purely by either the fast or slow mechanism. Thus, both forms of release can occur in isolation. These results imply that fast SVs are not derived from maturation of slow SVs, and vice versa. Our results are more compatible with the idea that fast and slow release comprise independent exocytosis mechanisms that operate in parallel. Fast and slow release share a requirement for UNC-13L, which provides a potential mechanism for coordinately regulating changes in these two release pathways.

Adjusting the contributions of fast and slow release will alter the duration of postsynaptic responses. This could have several important consequences for circuit function. Prolonged EPSCs will result in increased integration of synaptic inputs, which could alter the rate of postsynaptic spiking. Shifting the kinetics of postsynaptic responses could also alter spike timing-dependent plasticity. In sensory circuits, prolonged postsynaptic responses could alter the temporal integration of sensory inputs, and may alter the kinetics of sensory evoked responses. All these changes could have profound effects on circuit development, cognition, and behavior. Consistent with this idea, we recently showed that mutations

linked to Autism alter the kinetics of neurotransmitter release (*Hu et al., 2012*). For these reasons, there is significant interest in identifying molecular mechanisms that adjust the kinetics of neurotransmitter release.

In summary, we propose that the kinetics of neurotransmitter release are dictated by a protein code consisting of fast (UNC-10/RIM, UNC-13L) and slow (UNC-13L, UNC-13S, and Tomosyn) priming factors, which mediate fusion of proximal and distal SVs, respectively. All these molecules are conserved across phylogeny, and orthologous proteins are found at both NMJs and central synapses. Thus, we propose that the basic mechanisms governing the kinetics of neurotransmitter release are ancient, and will apply to many synapses. Consistent with this idea, a recent study showed that Munc13 proteins that lack C2A domains (bMunc13-2 and Munc13-3) are located more distally and mediate slow release at the Calyx of Held (*Chen et al., 2013*).

# Materials and methods

## Strains

Strain maintenance and genetic manipulation were performed as described (*Brenner, 1974*). Animals were cultivated at 20°C on agar nematode growth media seeded with OP50 bacteria. For electrophysiology, HB101 *Escherichia coli* was utilized. The following strains were utilized in this study:

Wild type, N2 bristol
KP6901 *unc-13(s69)*
KP6902 *unc-13(e1091)*
KP6903 *unc-13(e51)*
KP3707 *tom-1(nu468)*
KP3810 *tom-1(nu468) unc-13(s69)*
KP6904 *tom-1(nu468) unc-13(e1091)*
KP6905 *tom-1(nu468) unc-13(e51)*
KP7162 *tom-1(nu468);unc-10(md1117)*
KP5982 *acr-16(ok789)*
KP5984 *unc-29(x29)*
KP6893 nuEx1515 [P*snb-1*::UNC-13L];*unc-13(s69)*
KP6894 nuEx1516 [P*snb-1*::UNC-13L];*unc-13(e1091)*
KP6895 nuEx1517 [P*snb-1*::UNC-13L];*tom-1(nu468) unc-13(s69)*
KP6896 *nuIs46* [P*unc-13*::UNC-13S::GFP];*tom-1(nu468) unc-13(s69)*
KP6897 *nuIs486* [P*unc-129*::UNC-13L::mCherry]
KP6898 nuEx1518 [P*acr-2*::UNC-10::mCherry]
KP3928 *nuIs165* [P*unc-129*::UNC-10::GFP]
KP6899 *nuIs46* [P*unc-13*::UNC-13S::GFP];*unc-13(s69)*
KP7151 nuEx1590 [P*snb-1*::UNC-13SL];*unc-13(s69)*
KP7152 nuEx1591 [P*snb-1*::UNC-13ΔC2A];*unc-13(s69)*
KP7153 nuEx1592 [P*snb-1*::UNC-13R];*unc-13(s69)*
KP7154 nuEx1593 [P*snb-1*::UNC-13C2AR];*unc-13(s69)*
KP7155 nuEx1594 [P*snb-1*::UNC-13CaMR];*unc-13(s69)*
KP7156 nuEx1595 [P*snb-1*::UNC-13caMR(F574R)];*unc-13(s69)*
KP7157 nuEx1596 [P*unc-129*::UNC-13ΔC2A::mCherry]
KP7158 nuEx1597 [P*unc-129*::UNC-13C2AR::mCherry]
KP7159 nuEx1598 [P*unc-129*::UNC-13CaMR::mCherry]
KP7160 nuEx1599 [P*unc-129*::UNC-13S::mCherry]
KP7161 *nuIs490* [P*unc-129*::UNC-13R::mCherry]

## Constructs and transgenes

Transgenic strains were isolated by microinjection of various plasmids using either P*myo-2*::NLS-GFP (KP#1106) or P*myo-2*::NLS-mCherry (KP#1480) as coinjection markers. Integrated transgenes were obtained by UV irradiation of strains carrying extrachromosomal arrays. All integrated transgenes were out-crossed at least six times. UNC-13L and UNC-13S expression constructs contained full-length cDNAs, with the exceptions noted below. For UNC-13L, we utilized a full-length ZK524.2e cDNA

(spanning exons 1–31, excluding exon 14), whose expression as an SL1 spliced mRNA we confirmed by qPCR. The ZK524.2e cDNA corresponds to the LR mRNA previously described (*Kohn et al., 2000*). For UNC-13S, we confirmed expression of an SL1 spliced mRNA containing exons 14–31, which corresponds to the MR mRNA previously described (*Kohn et al., 2000*). We could not detect expression of ZK524.2c, a WormBase predicted short UNC-13 mRNA, by qPCR; therefore, we utilized the full-length MR cDNA for UNC-13S imaging and an MR minigene (*nuIs46*) (*Nurrish et al., 1999*) for electrophysiological recordings and behavior. Other UNC-13 constructs are as follows (all residue coordinates refer to the ZK524.2e protein sequence): ΔC2A (KP#1907), ZK524.2e cDNA lacking the C2A domain (residues 1–96); UNC-13R (KP#1908), cDNA containing exons 15–31 (residues 611–1816); C2AR (KP#1909), cDNA containing the C2A domain (residues 1–96) fused to the R domain; CaMR (KP#1910), and F574R (KP#1911), cDNAs containing the predicted CaM binding site (residues 556–610) or a mutated binding site fused to the R domain. For electrophysiological recordings, transgenes were expressed by the downstream *unc-13* promoter (UNC-13S rescue) or the *snb-1* promoter (all others). For imaging experiments, UNC-13 (c-terminal mCherry-tagged) and UNC-10 (*nuIs165*, c-terminal GFP tagged) (*Ch'ng et al., 2008*) transgenes were expressed in DA and DB neurons, using the *unc-129* promoter.

## Locomotion and behavior assays

Worm tracking and analysis were performed as previously described (*Dittman and Kaplan, 2008*) with minor modifications. Briefly, worms were reared at 20°C and moved to room temperature 30 min before imaging. Young adult animals were picked to agar plates with no bacterial lawn (30 worms per plate). Locomotion was analyzed 10 min after the worms were removed from food. 30 s videos of individual animals were captured at 5× magnification and 4 Hz frame rate on a Zeiss Discovery Stereomicroscope using Axiovision software. The center of mass was recorded for each animal on each video frame using the object tracking software in the Axiovision software. The trajectories were then analyzed using custom software written in Igor Pro 5.0 (Wavemetrics, Lake Oswego, OR).

## Fluorescence imaging

All quantitative imaging was done on a Zeiss Axioskop, using an Olympus PlanAPO 100 × 1.4 NA objective and a CoolSNAP HQ CCD camera (Roper, Trenton, NJ). Worms were immobilized with 30 mg/ml BDM (Sigma). Image stacks were captured and maximum intensity projections were obtained using Metamorph 7.1 software (Molecular Devices, Sunnyvale, CA). GFP fluorescence was normalized to the absolute mean fluorescence of 0.5-mm FluoSphere beads (Molecular Probes, Eugene, OR). For dorsal cord imaging, young adult worms, in which the dorsal cords were oriented toward the objective, were imaged in the region midway between the posterior gonad bend at the tail. Line scans of dorsal cord fluorescence were analyzed in Igor Pro (WaveMetrics) using custom-written software (*Dittman and Kaplan, 2006*).

## Electrophysiology

Electrophysiology was done on dissected adult *C. elegans* as previously described (*Richmond et al., 1999*). Worms were superfused in an extracellular solution containing 127 mM NaCl, 5 mM KCl, 26 mM $NaHCO_3$, 1.25 mM $NaH_2PO_4$, 20 mM glucose, 1 mM $CaCl_2$, and 4 mM $MgCl_2$, bubbled with 5% $CO_2$, 95% $O_2$ at 20°C. Whole-cell recordings were carried out at −60 mV using an internal solution containing 105 mM $CsCH_3SO_3$, 10 mM CsCl, 15 mM CsF, 4 mM $MgCl_2$, 5 mM EGTA, 0.25 mM $CaCl_2$, 10 mM HEPES, and 4 mM $Na_2ATP$, adjusted to pH 7.2 using CsOH. Under these conditions, we only observed endogenous acetylcholine EPSCs. For endogenous GABA IPSC recordings, the holding potential was 0 mV, at which we only observe GABAergic postsynaptic currents. All recording conditions were as described (*Hu et al., 2012*). Stimulus-evoked EPSCs were stimulated by placing a borosilicate pipette (5–10 μm) near the ventral nerve cord (one muscle distance from the recording pipette) and applying a 0.4 ms, 30 μA square pulse using a stimulus current generator (WPI). On average, this stimulus evoked ~85% of the charge evoked by hypertonic sucrose, suggesting that a large fraction of the primed pool was released. Statistical significance was determined using a two-tailed Student's *t*-test.

## Acknowledgements

We thank the *C. elegans* Genetics Stock Center for strains and reagents. We thank members of the Kaplan lab, Jihong Bai, and Jeremy Dittman for helpful discussions and critical comments on the manuscript.

# Additional information

## Funding

| Funder | Grant reference number | Author |
| --- | --- | --- |
| National Institutes of Health | R01 GM54728 | Joshua M Kaplan |

The funder had no role in study design, data collection and interpretation, or the decision to submit the work for publication.

## Author contributions

ZH, X-JT, Conception and design, Acquisition of data, Analysis and interpretation of data, Drafting or revising the article; JMK, Conception and design, Analysis and interpretation of data, Drafting or revising the article

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
