## [Decision Letter]

Thank you for sending your work entitled “UNC-13L, UNC-13S, and Tomosyn form a protein code for fast and slow neurotransmitter release” for consideration at *eLife*. Your article has been favorably evaluated by a Senior editor and 3 reviewers, one of whom is a member of our Board of Reviewing Editors.

The following individuals responsible for the peer review of your submission have agreed to reveal their identity: Graeme Davis (Reviewing editor) and Erik Jorgensen (peer reviewer).

The Reviewing editor and the other reviewers discussed their comments before we reached this decision, and the Reviewing editor has assembled the following comments to help you prepare a revised submission.

Hu et al. provide exciting new data on a few longstanding questions in synaptic research regarding heterogeneity of release-probability of synaptic vesicles. They use the *C. elegans* neuromuscular junction, which displays two clearly distinct kinetic components of release and exploit the relative ease of genetic manipulations in this system to provide important new information regarding two sources of heterogeneity: localization of vesicles relative to Ca-sources and intrinsic “willingness” of vesicles to fuse. They relate these aspects to the only two isoforms of the UNC-13 protein, which are present in *C. elegans*, suggesting roles of the C2A domain of UNC-13L in proper localization of vesicles relative to the dense projection and the calmodulin-binding domain of the same isoform in accelerating release. Furthermore, they describe an inhibitory/competitive role of Tomosyn, acting on UNC-13S mediated release. The data are of highest relevance for the understanding of synaptic function.

All of the reviewers consider this work high profile and appropriate for publication in *eLife*. No additional experiments are necessary prior to publication. However, several issues should be addressed in the text prior to resubmission and publication. In general the manuscript could be improved by re-organization. This would take considerable work and is not necessary for publication, but it is a suggestion that the authors might wish to consider. Two textual revisions are considered essential for publication, highlighted as #1 and #2 under “Essential revisions”.

Essential revisions:

1) The reviewers strongly suggest a more cautious interpretation of the CaM-domain data. In essence, does the CaM-domain increase or decrease sensitivity? A high Ca^++^ sensitivity of the diffusely distributed CamMR would result in prolonged, EGTA-sensitive release. A decreased Ca^++^ sensitivity, on the other hand, would produce short-lived EGTA-insensitive release, since release would be restricted to the peak-Ca, both temporally and spatially, since release probability is low and a high-power function of local [Ca^++^]. It should also be noted that the UNC-13L rescue has very low quantal content (Figure 3), as does the CAMR under EGTA. The final word regarding the mechanistic interpretation will require quantitative modeling. This, however, clearly goes beyond the scope of a single paper, since it requires knowledge about pool sizes and release probabilities, which may be difficult to quantitate. The conclusion that the CaM-binding domain makes fusion EGTA resistant would seem to be a safe conclusion.

2) The absence of detailed cartoons diagramming the constructs makes a careful reading of the manuscript difficult and, at times, confusing. It was not clear what the constructs looked like and how much of the surrounding region was included or whether there was a flexible linker. Given that supplements are now commonplace, it seems that DNA constructs can be described precisely, so that future researchers will be able to either repeat or interpret the experiments.

---

## [Author Response]

*1) The reviewers strongly suggest a more cautious interpretation of the CaM-domain data. In essence, does the CaM-domain increase or decrease sensitivity*?

We agree that it was unwise to speculate about how the CaM binding site alters the overall calcium sensitivity of release. As suggested, we deleted that speculation from our Discussion, which now reads: “Our results suggest that this calmodulin binding site also accelerates release. Thus, the faster kinetics of UNC-13L mediated release results from both tighter spatial coupling to calcium channels, and faster intrinsic kinetics of exocytosis.”

*2) The absence of detailed cartoons diagramming the constructs makes a careful reading of the manuscript difficult and, at times, confusing. It was not clear what the constructs looked like and how much of the surrounding region was included or whether there was a flexible linker. Given that supplements are now commonplace, it seems that DNA constructs can be described precisely, so that future researchers will be able to either repeat or interpret the experiments*.

As requested, we now provide a new figure, Figure 8—figure supplement 1, detailing all of the UNC-13 constructs utilized in our study.